# Vertical ocean heat redistribution sustaining sea-ice concentration trends in the Ross Sea

Olivier Lecomte[1], Hugues Goosse[1], Thierry Fichefet[1], Casimir de Lavergne[2], Antoine Barthélemy[1] & Violette Zunz[3]

Several processes have been hypothesized to explain the slight overall expansion of Antarctic sea ice over the satellite observation era, including externally forced changes in local winds or in the Southern Ocean's hydrological cycle, as well as internal climate variability. Here, we show the critical influence of an ocean–sea-ice feedback. Once initiated by an external perturbation, it may be sufficient to sustain the observed sea-ice expansion in the Ross Sea, the region with the largest and most significant expansion. We quantify the heat trapped at the base of the ocean mixed layer and demonstrate that it is of the same order of magnitude as the latent heat storage due to the long-term changes in sea-ice volume. The evidence thus suggests that the recent ice coverage increase in the Ross Sea could have been achieved through a reorganization of energy within the near-surface ice-ocean system.

[1] Université catholique de Louvain, Earth and Life Institute, Georges Lemaître Centre for Earth and Climate Research (UCL-ELI-TECLIM), Place Louis Pasteur 3, Bte L4.03.08, 1348 Louvain-la-Neuve, Belgium. [2] Sorbonne Universités (Université Pierre et Marie Curie Paris 6)-CNRS-IRD-MNHN, LOCEAN Laboratory, F-75005 Paris, France. [3] Vrije Universiteit Brussel (VUB), Earth System Science & Department of Geography, Pleinlaan 2, 1050 Brussels, Belgium. Correspondence and requests for materials should be addressed to O.L. (email: olivier.lecomte@uclouvain.be)

The overall positive trend in ice extent in the Southern Ocean since the late 1970s results from a strong increase in ice concentration (i.e., the local percentage area covered by ice) in the Ross and Weddell Seas, partly compensated by a decrease in the Bellingshausen Sea[1] (Fig. 1a). This pattern has partly been attributed to wind changes[2–4], potentially forced by the decrease in stratospheric ozone concentration and increased greenhouse gas concentrations[5, 6]. A second line of thought argues that a freshening of the upper ocean, due to either increased precipitation rates[7–9] or enhanced freshwater discharges from the melting Antarctic ice shelves[10, 11], has decreased the ocean surface density, reduced the convection, and enhanced the oceanic stratification. Changes in the Southern Ocean surface stratification may also be achieved through the northward export of freshwater associated with sea-ice advection[12, 13], as sea-ice forms in cold regions relatively close to the continent and releases freshwater farther north after transport by winds and ocean currents. As the Antarctic sea-ice zone is characterized by the presence of Circumpolar Deep Water (CDW) under the mixed layer, which is warm compared to the surface water, an increased stability due to surface freshening leads to smaller vertical heat fluxes toward the surface layer. This ultimately favors ice production and inhibits ice melt. The large upward oceanic heat flux to the ice in the Southern Ocean is indeed a central element in the sea-ice development and small variations of its magnitude have a large impact on the ice thickness and extent[14]. The role of changes in stratification is supported by several studies, showing in particular a surface freshening of the Ross Sea[15] and a circumpolar intensification of near-surface vertical salinity and temperature gradients[16].

Internal climate system variability may have played a strong role in the observed sea-ice trends[17–19]. In particular, changes in ocean stratification and vertical oceanic heat flux are not necessarily caused by an external perturbation in freshwater forcing. Internal ice-ocean processes can also lead to a redistribution of the salt content, horizontally or vertically, producing potentially large changes in stratification on decadal to multi-decadal time scales. In particular, a mechanism related to the interplay between interannual variations and the seasonal cycle can engender a net vertical transport of salt in response to an initial perturbation, altering the near-surface stratification and amplifying initial changes[20]. Specifically, the mixed layer depth (MLD) under sea ice has a marked seasonality, with large values in winter associated with convection when sea ice forms and releases brine, and small ones in summer when sea ice melts and freshwater stabilizes the upper ocean column. The salt rejected during ice production in winter is thus mixed over a deep layer, while the freshwater due to melting is incorporated into a shallow mixed layer. In an equilibrium climate, if we neglect for simplicity the potentially important contribution of horizontal transport, the situation is stable because the same amount of salt transported to depth one year is entrained into the mixed layer the following year. However, if ice formation is particularly large during one or a few years, the brine released can be transported downward to deeper layers and not incorporated back into the mixed layer of the subsequent winters. This leads to a decrease in surface salinity, a stronger stratification, a shallower mixed layer and thus reduced vertical oceanic heat fluxes. Consequently, heat is trapped at depth in response to the larger sea-ice formation, leading to a positive feedback henceforth referred to as ice-coverage–ocean-heat-storage feedback.

Hereunder, we show the signature of this feedback and quantify the heat trapped at the base of the ocean mixed layer in the Ross Sea, where long-term positive sea-ice trends are observed and simulated by a model. We demonstrate that the heat gain is of the same order of magnitude as the latent heat storage due to the associated changes in sea-ice volume, indicating that the trends may be sustained via a vertical redistribution of energy in the near-surface ice-ocean system. Conversely, this implies that enough energy is available at depth to melt the extra ice formed during the last few decades, should the mechanism be reversed.

## Results

**The Ross Sea**. The Ross Sea sector is the region hosting most $(13.7 \pm 3.6 \ 10^3 \ \text{km}^2 \ \text{yr}^{-1})$ of the overall sea-ice extent increase around Antarctica $(17.5 \pm 4.1 \ 10^3 \ \text{km}^2 \ \text{yr}^{-1})$ over the last three decades[1]. Here, we study the changes in Antarctic sea-ice coverage using both the OSISAF[21] sea-ice concentration satellite retrievals over 1979–2014 and sea-ice fields from a simulation performed with the global ocean–sea-ice model NEMO-LIM3.6[22, 23], driven by atmospheric reanalyses

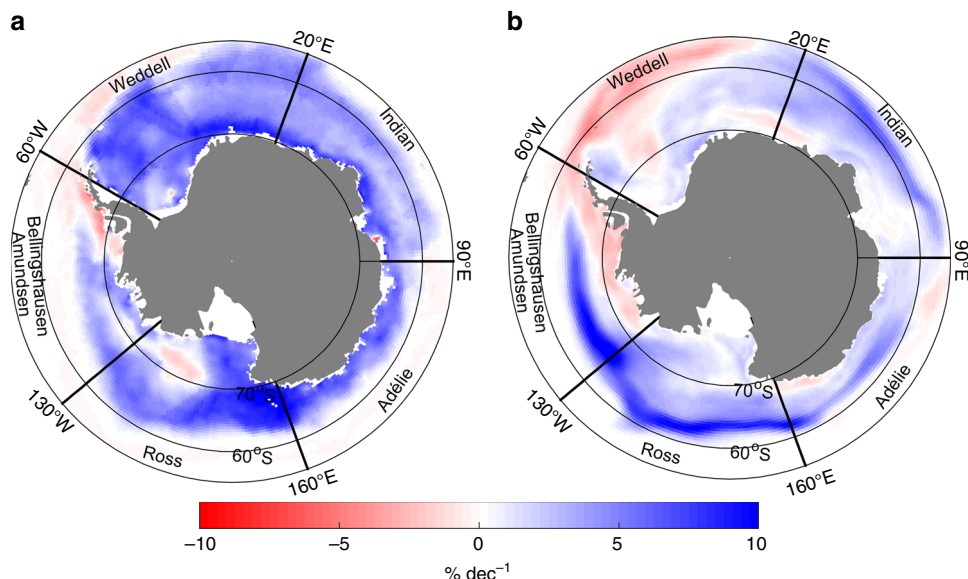

**Fig. 1** Observed and simulated Antarctic sea-ice concentration trends. Observed[21] **a** and NEMO-LIM3.6-based **b** annual trends in Antarctic sea-ice concentration, computed over 1979–2014 and 1979–2013, respectively. All sectors of the Southern Ocean are defined as in a previous study[37]

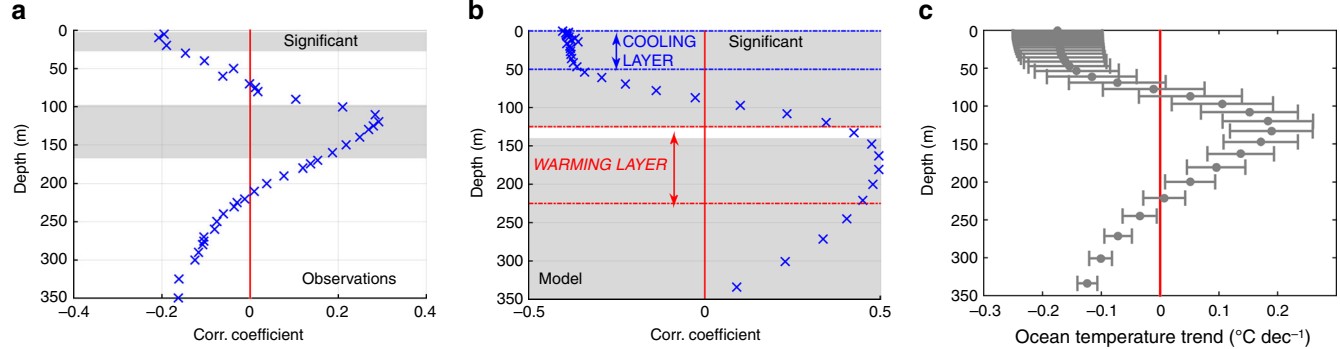

**Fig. 2** Correlation coefficients between ice concentration and depth-dependent ocean temperature trends in the Ross Sea. Observation-based correlations **a** are computed from the OSISAF[21] sea-ice concentration satellite product and ocean temperature profiles from the BLUELink Ocean Archive[27]. The same diagnostic is applied to a simulation performed with the ocean–sea-ice model NEMO-LIM3.6 (**b**, see Methods section). **c** depicts the mean depth-dependent ocean temperature trends (model-based) over Ross Sea areas, where ice concentration trends are positive and considered significant, i.e., larger than 3% dec$^{-1}$, the 95% confidence interval for ice concentration trends in the model, and observations. The regions associated with **b** and **c** are thus slightly different, with **b** including areas with deeper MLD. At each depth, the *horizontal bar* represents the standard deviation around the spatially averaged trends. Trends are computed over 1979–2013 and 1979–2014 for the model and observations, respectively. *Gray* areas show the depths at which the correlations are significant at the 95% confidence level. Below 220 m, **a** and **c** show negative correlations and trends that are probably due to processes unrelated to sea-ice trends, hence not further discussed in this study

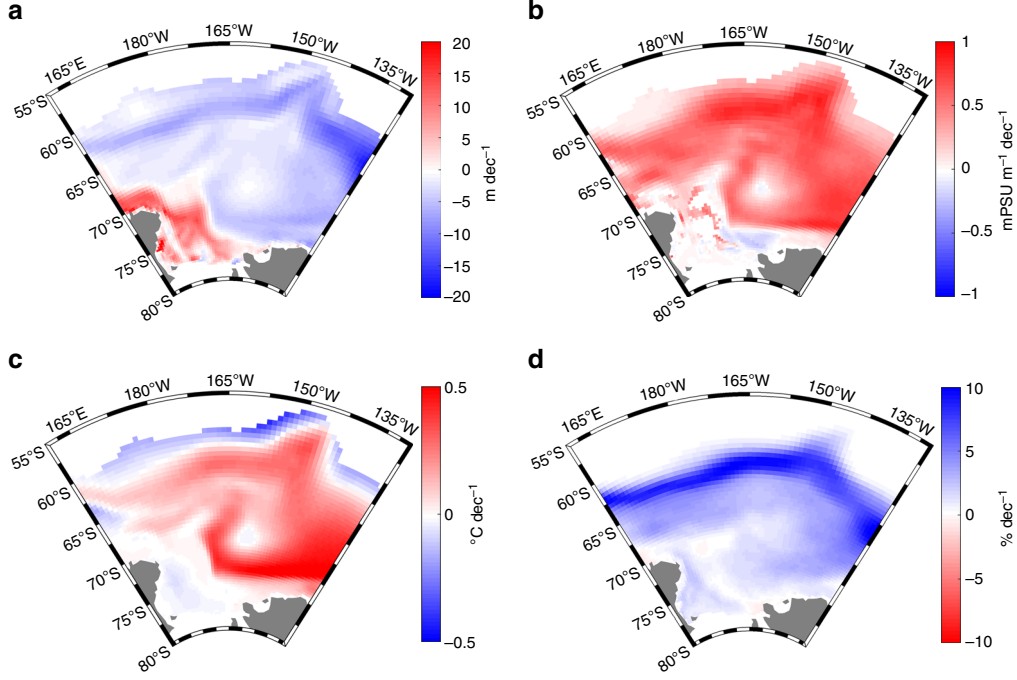

**Fig. 3** 1979–2013 model trends in upper ocean properties in the Ross Sea. 1979–2013 model trends in **a** mixed layer depth, **b** near-surface vertical salinity gradient, **c** ocean temperature at depth, and **d** sea-ice concentration in the sea-ice-covered region of the Ross Sea. In **c**, temperature trends are shown at the depth at which they are maximum, within the upper 300 m. Except in the southernmost Ross Sea where MLD increases **a**, the depth of this maximum is always greater than the local mean MLD (Supplementary Fig. 3). This depth is also used as the lower boundary of the column over which salinity gradients of **b** are computed

(over 1979–2013, see Methods section). Except for the outer Weddell and inner Amundsen Seas, this simulation reproduces reasonably well the observed sea-ice concentration trends (Fig. 1) and, in particular, exhibits the largest positive trends in the Ross Sea. Fully coupled climate models fail in general to simulate Antarctic sea-ice trends correctly[18, 24, 25], but the Ross Sea is the region where the mechanisms behind sea-ice trends are the most poorly understood and where the disagreements between climate simulations and observations are the largest[26]. Hereafter, we focus on the Ross Sea (our main diagnostic for the other regions is presented in Supplementary Figs. 1 and 2), as understanding the processes driving sea-ice changes in this region appears key to determining the origin of the slight circumpolar ice extent increase.

**Signature of the ice-coverage–ocean-heat-storage feedback**. The role of oceanic vertical heat fluxes is investigated based on observational ocean temperature profiles from the BLUELink Ocean Archive[27] (1979–2014), and on ocean outputs from the

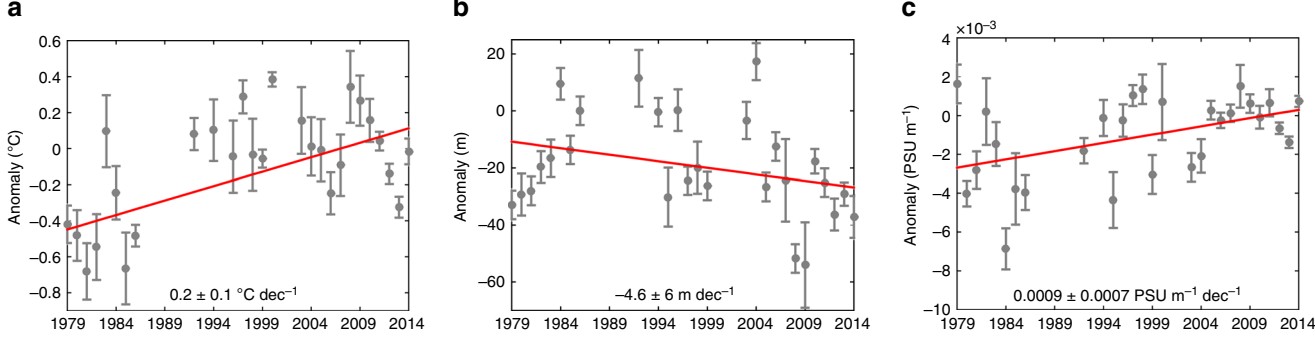

**Fig. 4** 1979–2014 observed annual time series and trends in upper ocean properties in the Ross Sea. 1979–2014 annual time series and trends (from in situ observations[27]) in **a** 100–160 m mean ocean temperature, **b** mixed layer depth, and **c** salinity gradient across the top 130 m, on average over the Ross Sea areas where observed ice concentration trends are larger than 3% dec⁻¹ (typical interval of confidence for ice concentration trends). Annual means (± one standard error) are shown in *gray*. Linear trends are shown as *red lines* and given with their 95% confidence interval at the bottom of each panel

model simulation. Compared to in situ observations, model results have the advantage of providing a complete spatial and temporal coverage. We evaluate the spatial correlation between trends in ice concentration and trends in ocean temperature at various depths (Fig. 2). All trends are evaluated from annual mean data over the full period length (see Methods section). A map showing the spatial distribution of the ocean temperature profiles used to compute the observation-based correlation is given in Supplementary Fig. 1. As expected, ice concentration and sea surface temperature trends are negatively correlated, meaning that an increase (decrease) in ice concentration is associated with a surface cooling (warming). Nonetheless, the correlation becomes positive as depth increases and reaches a maximum at a depth of nearly 120 m in observations and 150 m in the model: this entails that an increase (decrease) in ice concentration is also associated with heat gain (loss) at depth. Both peaks in positive correlation (from observational and model data sets) are significant at the 95% confidence level, despite weaker correlation values in the observations. In addition, those peaks occur at comparable depths, which in the model corresponds approximately to the depth of the winter mixed layer, averaged over all the seasonally ice-covered grid cells of the Ross Sea and over 1979–2013. Supplementary Figures 1 and 2 display these same correlations between ice concentration trends and ocean temperature trends applied to all sectors of the Southern Ocean. The significance of trends and subsequently of correlations is mainly conditioned by the number of records available. Correlations are thus more robust for model results than for observations. Over the last three decades, only the Ross Sea exhibits a distinct positive maximum in correlation at depth.

We argue that this positive correlation at depth is the signature of the ice-coverage–ocean-heat-storage feedback whereby increased sea-ice coverage conduces to, and is amplified by, a salinity-related strengthening of the stratification and a mixed layer shoaling. Indeed, the model simulation shows that, in regions of significant positive sea-ice concentration trends (Fig. 3d), increased stability (Fig. 3b) and reduced MLDs (Fig. 3a) act to weaken the upward oceanic heat transfer, causing heat gain at and below the base of the mixed layer (Figs. 2c and 3c). Although the number of observational temperature profiles available in the seasonally ice-covered Ross Sea is too limited to highlight a clear geographical match between strong trends in MLD, subsurface temperature and ice concentration, trend correlations at the scale of the Ross Sea concur with the proposed mechanism. Indeed, the vertical structure of the correlations (Fig. 2a) shows strong similarity with the simulated profile of temperature trends in the region of large sea-ice increase (Fig. 2c), which is also the region where the observational coverage is

densest (Supplementary Fig. 1). Further, Fig. 4 shows that on average over the region where positive ice concentration trends are the most significant, a simultaneous shoaling of the mixed layer, increase in surface stratification, and warming at depth seem to occur in observations too. Though these trends are highly uncertain, they do not invalidate our hypothesis and suggest a behavior broadly consistent with that simulated.

**A two-way process not specific to recent decades.** The correlation pattern observed for the Ross Sea only over the period 1979–2013 is found in all but one sector and for the whole Southern Ocean in model results over 1952–1978 (Supplementary Fig. 5). The relationship between trends in sea-ice concentration and ocean temperature is consequently not specific to the 1979–2013 period and thus does not seem to be driven by the climate change of the past three decades[28]. The Weddell, Indian, and Bellingshausen–Amundsen sectors, in particular, present a correlation profile over 1952–1978 analogous to that found for the Ross Sea during 1979–2013. However, the sign of the trends is reversed. Supplementary Figure 6 shows a deepening of the mixed layer and cooling of subsurface waters co-occuring with negative sea-ice concentration trends and stratification weakening over much of the Antarctic sea-ice zone (note that the situation is different at the centre of the Ross gyre due to a shallow MLD anomaly, blurring the feedback signature when averaging over the Ross sector over that period). A cooling at depth is thus associated with a decay of the sea-ice cover. This indicates that the reorganization of energy within the ocean–sea-ice system may occur in both ways, provided that an initial perturbation enables its establishment over a longer, multi-decadal time scale.

**Relationship with seasonality.** The ice-coverage–ocean-heat-storage positive feedback is intrinsically related to the seasonal cycle of the mixed layer but it applies to decadal trends of annual mean ice concentration. It favors the production of ice and inhibits its melting when heat is stored at depth; conversely, it inhibits production and favors melting when heat stored at depth is released. Hence, it cannot explain alone opposite trends across seasons. In the Ross Sea, 1979–2013 sea-ice trends are coherent across seasons (Supplementary Fig. 4), leading to strong annual mean trends and a clear feedback signature. In other sectors, the trends over 1979–2013 vary between seasons, leading to smaller trends in annual average, and suggesting that other processes with out-of-phase impacts across seasons are dominant[29]. Furthermore, the insignificant positive correlations from 0 to 50 m in the Bellingshausen–Amundsen sector in observations (Supplementary Fig. 1) may also be explained by

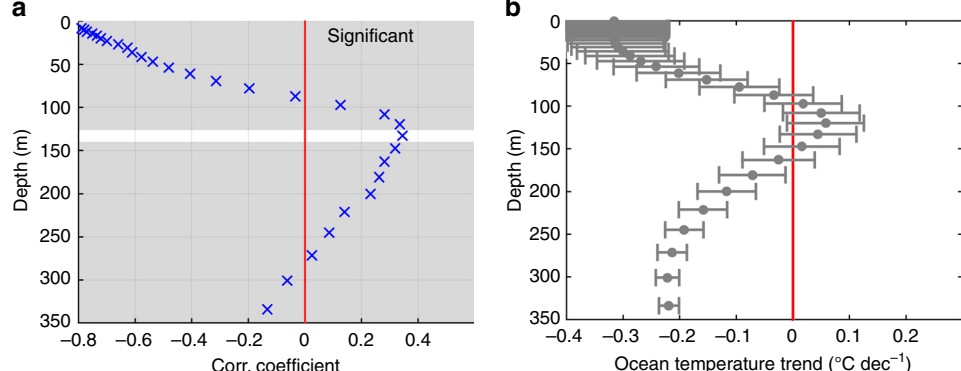

**Fig. 5** Relationship between trends in ice concentration and ocean temperature at depth for the model driven by climatological winds. **a** Correlation between 1979–2013 trends in ice concentration and ocean temperature at depth for the model driven by climatological winds. *Gray areas* show the depths at which the correlations are significant at the 95% confidence level. **b** Mean depth-dependent ocean temperature trends (model-based, same simulation) over Ross Sea areas where ice concentration trends are positive and considered significant, i.e., larger than 3% dec$^{-1}$, the typical interval of confidence for ice concentration trends. At each depth, the *horizontal bar* represents standard deviation around the spatially averaged trends

seasonality. If winter and summer trends in ice concentration have opposite signs, the correlations computed from annual values of sea-ice concentration are difficult to interpret and may even become meaningless. This is why a statistically significant maximum in positive correlation between ice concentration and ocean subsurface temperature trends is found only in the Ross Sea, where summer and winter trends are consistent. Note that although changes are expected to have the same sign all year round if the ice-coverage–heat-storage feedback is the dominant process, they need not have the same magnitude in all seasons. Furthermore, the feedback may be triggered by processes occurring during a specific season, and is thus compatible with the suggestion that sea-ice positive trends in the Ross Sea are initiated primarily in the melt season[30]. In short, each single season may have a specific influence on the feedback intensity and its signature, but as it establishes over long-time scales and is the result of the interplay between mechanisms happening in different seasons, the role of each season cannot be isolated.

**Quantifying the vertical heat redistribution**. In order to quantify the impact of the ice-coverage–ocean-heat-storage feedback on the heat balance of the Ross Sea in the model, we have computed the ratio $R_H$ defined as (see Methods section) the sum of the latent heat changes associated with the sea-ice volume trends and the cooling at the surface over the concomitant heat content changes at depth. As illustrated in Fig. 2b, a cooling surface layer and a warming subsurface layer, characterized by the peaks in correlation, are defined as the top 50 m and the 125–225 m layers of the ocean, respectively. These layers are used to calculate ocean heat content changes entering $R_H$. On average over the Ross Sea in our 1979–2013 model simulation, $R_H$ approximately equals 1. Although the value of this coefficient is sensitive to the way the cooling and warming layers are defined (ranging from 0.5 to 3 for extreme definitions of those layers), this indicates that the heat losses in the ocean surface–sea-ice system and the heat gain at depth have the same order of magnitude. Unfortunately, the equivalent of $R_H$ cannot be evaluated from observations because of the lack of ice thickness measurements. Additional temperature measurements below the ice pack, especially in the poorly sampled inner Ross Sea, would also help to better evaluate $R_H$ and to improve estimates of the correlation with ice concentration. It is also possible to compute an equivalent to $R_H$ for the classical negative feedback[14] in which an initial ice growth is partly compensated by a larger ocean-ice heat flux because of a mixed layer deepening and entrainment of warmer

water. Nevertheless, as the time scale and processes are different for the decadal ice-coverage–ocean-heat-storage feedback discussed here, it may not be straightforward to make a connection between them.

**Discussion**
The mechanism described above seems to explain well the observed and simulated changes in the Ross Sea. Observations are too scarce to unequivocally identify the causes of changes in MLD, but it is possible to test, in model results at least, if alternative mechanisms are also at play. Our simulation is performed using climatological run-off and precipitation forcing fields that were detrended[31]. Furthermore, the freshwater flux from icebergs and ice shelf melting is constant over the whole simulation. The enhancement of upper ocean stratification in Ross Sea regions associated with large ice concentration increases can thus not be due to a trend in precipitation or continental melt water input.

A second hypothesis is that the subsurface temperature increase is primarily due to an independent warming and shoaling of CDW[32]. This seems unlikely, first because a robust correlation of temperature trends at depth with ice concentration trends would not be expected if those two signals were, in fact, independent. Furthermore, Fig. 2c shows that the subsurface warming occurring under large sea-ice concentration increases is concentrated within a bulge around 150 m depth, with a cooling below 200 m. An upward shift of CDW would instead be associated with a warming on a larger depth range, since the temperature of CDW reaches its maximum at 330 m in this area in the model. The shoaling of CDW is however more likely to explain the positive correlations between model trends in ice concentration and ocean temperature below 300 m in the Indian, Adélie, and Bellingshausen–Amundsen sectors (Supplementary Fig. 2), especially as they are not seen over 1952–1978 (Supplementary Fig. 5).

A third hypothesis is that the trends are simply forced by wind trends, which would induce a higher northward advection of sea ice and then freshwater. The important role of winds and sea-ice transport in many recent changes has been demonstrated in several studies[3, 12, 13]. However, if winds and freshwater transport most probably play a role in initiating and amplifying the mechanism, our experiments show that their trends are not the primary cause of the simulated decadal sea-ice trends. The main evidence is provided in Fig. 5. The correlation between ice concentration and ocean temperature trends are shown in the

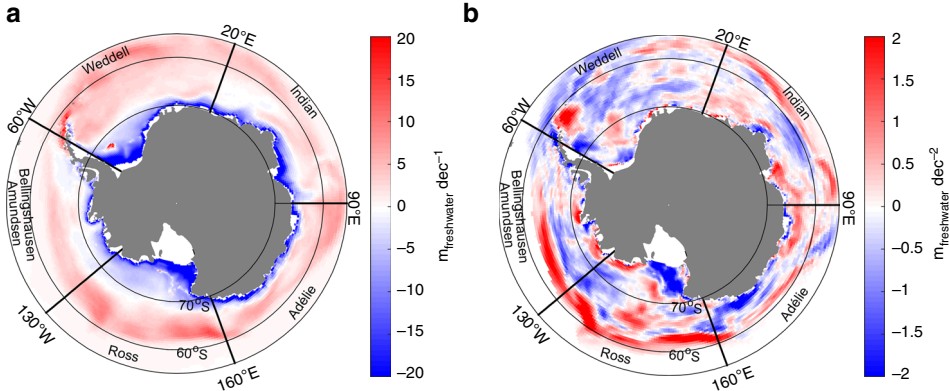

**Fig. 6** 1979–2013 freshwater equivalent flux resulting from sea-ice formation and melting in the model. **a** Mean flux in the ice-covered region of the Southern Ocean, as simulated by NEMO-LIM3.6. **b** Trends in freshwater equivalent flux over 1979–2013

Ross Sea for a simulation similar to the one described in Methods section, except that the model is forced by climatological winds over 1958–2013, all other forcing fields remaining unchanged (i.e., with interannual variations). Although this simulation has no trend in wind stress, it still has a positive trend in ice concentration and extent in this sector, and the signature of the heat trapping at depth associated with positive sea-ice concentration changes appears in the correlations. In the initial simulation (with the original wind forcing), this signature is stronger, especially when looking at depth-dependent temperature trends (Figs. 2c and 5b) because the wind strengthens the ice-ocean feedback in this experiment by favoring larger ice concentration trends and hence larger heat trapping at depth. Changes in MLD analyzed here are thus not a mere response to surface wind trends.

Advection still contributes to the trend and its effect is likely to be most prominent close to the ice edge. Trends are indeed large there in our results, suggesting a contribution of wind-induced changes, but positive ice concentration trends are also associated with a warming at depth in other regions such as the eastern inner Ross Sea. More generally, the trends in freshwater flux at the ocean surface resulting from sea-ice formation and melting (Fig. 6) show no general relationship with the trends in ice concentration and MLD. Figure 6a displays the average of this freshwater flux over 1979–2013. Regions in blue, along the coasts of Antarctica, exhibit strong deficits in freshwater due to the large local ice growth rates[33] in winter. Virtually everywhere else, areas in red illustrate the positive net balance in freshwater at the ocean surface, ensuing from the summer melt of ice advected away from the coast. Figure 6b then shows that, despite some apparent correspondence with sea-ice concentration and MLD trends (Figs. 1b and 3) in the Indian, Ross, and Amundsen sectors at the ice edge, the trends in freshwater flux are much more variable spatially, and their actual correlation with ice concentration trends is very low ($\approx 0.05$ for the whole Southern Ocean, $\approx 0.15$ for the Ross Sea). This suggests that the long-term mixed layer shoaling is not, or not solely, attributable to a net local import of freshwater because of anomalous ice transport.

Nonetheless, the northward export of freshwater associated with sea-ice advection may influence the feedback strength by way of an additional indirect dilution effect. For instance, let us assume that an event induces a year-round shoaling of the mixed layer, all else remaining equal, to simplify. In regions of net annual surface freshwater gain (that is, almost everywhere in the Southern Ocean except along the Antarctic coasts[13]), the shallower mixed layer results in a stronger dilution of the mixed layer salt and thereby in a stronger surface salinity drop. Even without any further reductions in MLD, nonetheless expected, the

shallower mixed layer can then lead to a steady surface freshening and stratification increase. In our simulation, Ross Sea trends in ice concentration do not correlate well in space with trends in surface freshwater input (Fig. 6b), but they do correlate better with the mean state of this flux (Fig. 6a, correlation of $\approx 0.4$). Additionally, Supplementary Fig. 7 indicates that, on average over the seasonally ice-covered area of the Ross Sea, sea surface salinity decreases over 1979–2013, while the amplitude of its seasonal cycle increases over the same period. These trends are consistent with the presence of a dilution effect amplifying the ice-coverage–ocean-heat-storage feedback. The effect may also explain why the feedback operates more clearly in some open ocean regions characterized by a net surface freshwater gain.

Hence, vertical correlation profiles between ice concentration and ocean temperature trends (Fig. 2), combined to the heat storage ratio $R_H$, substantiate the ability of an internal ice-ocean feedback to sustain multi-decadal sea-ice trends. In particular, the new diagnostics proposed here provide a novel tool for the process-oriented evaluation of climate models. Our analysis could be used in broad inter-model comparisons to examine (1) whether or not models can represent the positive ice-coverage–ocean-heat-storage feedback, (2) the depth of its ocean temperature signature, and (3) its intensity through the heat storage coefficient $R_H$ we suggest here.

The results indicate that the reorganization of heat within the upper-ocean–sea-ice column is by itself sufficient to sustain the recent sea-ice expansion in the Ross Sea, which accounts for most of the circumpolar expansion. The model analysis suggests that the proposed feedback may be amplified by wind trends and by a dilution effect related to freshwater export into regions experiencing mixed layer shoaling, but it may also occur without them. Our results further imply that the heat trapped in the warming layer of the Ross Sea represents enough energy to melt the ice formed previously, should the process be reversed and the heat released at the surface. Finally, the positive feedback described here may reconcile the various hypotheses existing to explain the recent Antarctic sea-ice trends: externally forced changes in ice dynamics (through changing wind regimes) or ocean freshwater balance (owing to increased precipitation or ice-sheet melting) may have provided the initial perturbation necessary to trigger an internal mechanism in the ocean–sea-ice system that sustains further sea-ice changes.

## Methods

**Observations.** The analyses are based on the ocean temperature and salinity profiles from the BLUELink Ocean Archive[27], and an estimate of the seasonal cycle (CARS-Argo climatology, period 2004–2014) of temperature ($T$), salinity ($S$) and MLD. Specifically, this BOA-Argo-based climatology provides coefficients for the

first two harmonics of seasonal variations, allowing computation of climatological values for any day of year. We first preselect temperature and salinity profiles with a minimum 25 m resolution in the top 350 m of the ocean and interpolate them to ensure complete coverage over this same depth range. Then, for each temperature or salinity record, a vertical profile of anomalies is obtained by subtracting the climatological estimate at the day of year and location of the observation. In order to obtain MLD from $T$ and $S$ profiles, preselected $T$ and $S$ profiles are paired using a criteria based on distance ($<5$ km), time ($<25$ h), and station number correspondence. From each $T$–$S$ profile pair, we compute the associated MLD value following the $T$–$S$ threshold criterion[34] employed for the MLD climatology: the minimum of the depths $z1$, $z2$ satisfying $|T(z1) - T(10m)| = 0.2$, $|S(z2) - S(10m)| = 0.03$. An MLD anomaly is again obtained by subtracting the climatological estimate at the day of year and location of the observation.

Working with anomalies is critical here to sidestep the important issue of seasonal biases in the sampling of the ocean. Profiles of $T$–$S$ anomalies and MLD anomalies are then averaged annually on each cell of a regular $1° \times 1°$ grid, extending from 40 to 80ºS, using all records available within the corresponding grid cell every year. Linear regressions (over 1979–2014) can then be applied to the various fields independently. As the focus of our study is on multi-decadal time scales, only grid cells including records spread over at least 25 years were considered in order to compute trends.

We use sea-ice concentration satellite data from the OSISAF[21] Global Sea Ice Concentration Climate Data Records (1978–2015). These data were interpolated on the same grid as the one for ocean observations, and long-term trends in yearly-averaged ice concentration were computed in each grid cell.

All observation-based trends are computed over 1979–2014. Indeed, the reprocessed data set we use for ice concentration covers up to April 2015 only. After this date, another near real-time OSISAF product exists (continuous reprocessing offline product), but it is not fully consistent with the reprocessed data set, which makes it unsuitable for our application.

**Model fields**. NEMO-LIM3.6 is a state-of-the-art global ocean–sea-ice coupled model[22, 23]. The standard simulation analyzed here was performed over 1958–2013 using DFS5.2 (DRAKKAR Forcing Sets, distributed with the NEMO model) atmospheric forcing fields[31]. An additional sensitivity experiment was conducted in order to assess the role of winds in the presented mechanism (Fig. 5). The experimental set-up for this simulation was the same as for the standard simulation, except climatological winds were computed based on DFS5.2 and used instead of the initial forcing fields. Selected daily outputs from these simulations include sea-ice concentration and thickness, ocean temperature, salinity and MLD, and total freshwater fluxes associated to sea-ice formation and melting. We first averaged these outputs to produce annual fields, from which we calculate the 1979–2013 trends in all variables on the model native grid (for all sea-ice-covered grid cells). As the DFS5.2 forcing set starts in 1958, a simulation using the model in a former configuration (described and evaluated previously[35]) was performed to study the changes over the period 1952–1978 (Supplementary Figs. 5 and 6).

**Diagnostics**. In order to produce the correlations displayed in Fig. 2, the following steps were applied independently to the observational and model data sets. Sea-ice concentration and ocean temperature trends were first calculated from annual data and only the statistically significant ones were considered for correlation calculations. Regarding model trends, significance was assessed by testing the null hypothesis with a Student's $t$-test, taking into account a reduced sample size due to lag-1 autocorrelation[36]. For observation-based data, however, a simple test without adjustment for autocorrelation was performed. The reason is that the lag-1 autocorrelation of temporally sparse data is often meaningless owing to the very small number of records it is based on. The spatial correlation coefficient between the distribution of ice concentration trends and ocean temperature trends was then computed at each ocean vertical level and presented for the first 350 m below the sea surface. In order to assess the impact of spatial autocorrelation in the data on these results, we performed a simple test. We reproduced the same diagnostics as in Fig. 2b after reducing the sample size at each vertical level by a factor of 9 (we selected only one out of every nine grid cells in the model data, assuming that any cell is correlated with its nearest neighbors in every directions). Doing so provides qualitatively the same results as Fig. 2b. For observations, only grid cells where both sea-ice trend and ocean temperature trend estimates are available were retained to compute these correlations. In Fig. 2, the vertical levels at which these correlations are significant ($p$-value $\leq 0.05$) are also shown in gray.

The heat storage ratio $R_H$, used to compare the amount of energy stored in subsurface layer to the heat losses associated with sea-ice volume changes and the ocean surface cooling, was defined as:

$$R_H = \frac{\rho_{ice} L \Delta v_{ice} - \rho_{sw} c_{sw} \Delta T_{cl} dz_{cl}}{\rho_{sw} c_{sw} \Delta T_{wl} dz_{wl}} \qquad (1)$$

where $\rho_{ice} = 917$ kg m$^{-3}$, $L = 334 \times 10^3$ J kg$^{-1}$, $\rho_{sw} = 1025$ kg m$^{-3}$, and $c_{sw} = 4 \times 10^3$ J kg$^{-1}$ K$^{-1}$ are the selected values for sea-ice density, the pure ice latent heat of fusion, the seawater density, and specific heat capacity, respectively. $\Delta v_{ice}$ represents the ice volume change per unit area over 1979–2013 (i.e., the ice volume trend over that period multiplied by 35 years) in the considered grid cell, $\Delta T_{cl}$ and

$\Delta T_{wl}$ are the temperature changes in the cooling and warming layers over the same period, and $dz_{cl}$ and $dz_{wl}$ are the thicknesses of the associated layers (Fig. 2). Therefore, this coefficient relates the heat trapped at depth to the sum of the latent heat associated with sea-ice changes and the heat loss of the ocean surface layer. $R_H$ was first computed in each model grid cell and was finally spatially averaged over the Ross Sea sector.

**Data availability**. All observational data supporting the findings of this study are publicly available online. BOA hydrographic data are available at https://researchdata.ands.org.au/bluelink-ocean-archive/692138?source=suggested_datasets, and the OSISAF sea-ice concentration satellite product at http://osisaf.met.no. The MATLAB code produced in order to perform all the data processing and analyses of this study is available upon request to the authors.

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

## Acknowledgements

We thank J. Dunn and D. Monselesan for providing and documenting the BOA data. Computational resources for this simulation have been provided by the supercomputing facilities of the Université catholique de Louvain (CISM/UCL) and the Consortium des Equipements de Calcul Intensif en Fédération Wallonie Bruxelles (CECI) funded by the Fonds de la Recherche Scientifique de Belgique (FRS-FNRS). O.L. and H.G. are, respectively, Postdoctoral Researcher and Research Director within the Fonds National de la Recherche Scientifique (F.R.S.-FNRS-Belgium). This work was supported by FNRS Research Project "Amélioration de la représentation de la glace de mer antarctique dans les modèles climatiques grâce à une meilleure compréhension des processus gouvernant son état moyen et sa variabilité", under Grant Agreement no. T.0007.14 and by Belgian Research Action through Interdisciplinary Networks—BRAIN-be—Belgian Science Policy Office, Grant Agreement no. BR/165/A2/Mass2Ant: "East Antarctic surface mass balance in the Anthropocene: observations and multiscale modeling (Mass2Ant)".

## Author contributions

All authors shared responsibility for writing the manuscript. O.L. analyzed the observational data, model output, and assembled the results. H.G. conceived and supervised the study. T.F. co-supervised the study. C.d.L. prepared the observational hydrographic data set and contributed to the interpretation of the results in all aspects of the analysis. A.B. designed the two main model experiments and V.Z. was involved in conceiving the correlation diagnostic proposed in the study.

## Additional information

**Competing interests:** The authors declare no competing financial interests.

