## [Peer Review File · Nature Communications]

Reviewers' comments:

Reviewer #1 (Remarks to the Author):

Review of

Antarctic sea-ice concentration trends sustained by vertical ocean heat redistribution
by Lecomte, Goose, Fichfet, de Lavergne, Barthelemy and Zunz

The authors use a comparison of an observational product and a model output to proof their hypothesis about the sea-ice extension origin.

The Manuscript is well written, and the findings are novel, though the authors do not test their hypothesis with different parameters or validate their results of modeled trends in ocean properties (e.g. Fig 3) against existing observational products, which are cited in their references. Further more the manuscript does need a more detailed description of the methods and data handling steps performed and an error analysis. Finally a discussion about the robustness of the results would be helpful, e.g. what other processes like the warming and rising of circumpolar deep water, could create a similar signal?

For a 2nd round of review, line numbers would be helpful,

In more detail:

Abstract, first sentence – there are quite strong regional differences, as the authors state themselves in the main text, this first sentence is to misleading, further more the reference to 'global warming' here is no appropriate, since to my knowledge it is not proven yet that the extended sea-ice in the southern ocean is linked to global warming.

Stating that the Ross Sea is the major contributor of sea-ice expansion needs a reference or rephrasing, sea ice drift patterns are not always simple to analyze or to be budgeted.

1st sentence main text. 'ice concentration' - please make clear throughout the manuscript if you talk about concentration within a given, 'fixed' area, maximum sea ice extent (which concentration threshold) or sea ice area (subtracted open areas, areas with 50% coverage only counting as 50%...).

So far it is not always obvious to the reader what the authors do refer to.

reference 19, which shows the Winter Water warming also show a freshening of Winter Water – this would contradict your hypothesis of vertical ocean heat transfer, since it would lead to a salinification, wouldn't it?

You state that in an equilibrium climate ... salt rejected at depth one year is entrained again in the mixed layer the following year. – I doubt this to be true for the majority of the southern ocean due to pronounced zonal currents, westwards in the north, eastwards near the coasts. Please justify your assumption.

Concerning your data set, the Bluelink Ocean Archive, I wonder why you restrict the analysis to 2013, since recent years significant amount of floats were available in your area of interest. – furthermore the BlueLink Ocean Archive is just binning data within a one degree grid cell. This method is not taking the current structures, bathymetry or data distribution in the southern ocean into account. Please justify the use of this data set in favor of the many other possible data sources? In particular on page 8 you state the availability of argo float and elephant seal data – this is publically available and can be used for your analysis!

page 4 first paragraph: in the sentence 2 you contradict your statement from the abstract, where you emphasize that models fail to reproduce this. So which of these 2 sentences is correct?
Last sentence, same paragraph: what is a 'reasonable' number of observations, reasonable is

spatial or temporal aspect? please rephrase that sentence.

page4 2nd paragraph: Please be more precise, which area, and grid cells do you compare, how do you compute the correlation, using monthly? Annual data? How do you handle data gaps that are common in observational data sets and how do you deal with the different temporal data coverage for each grid cell. Do you use the model only at the same grid cells as observations do exist? Last with all the grid cells in the Ross Sea, you provide just one correlation curve, please add error bars to see the spread within the Ross Sea.

page4 last paragraph: please guide the reader more thoroughly through your steps of conclusions. I found it hard to follow your chain of arguments.

page 5 1st paragraph: you state that the heat trapping is identified here in model and observation, please proof this hypothesis more carefully, e.g. by comparing salinity from observations in an identical way. For the limited region and limited data of the area, you need to back this bold statement, in particular because you state that other areas behave different.

page 8 last paragraph: 'clearly' – please rephrase. – please consider rewriting the last paragraph, to explain the possible shortcomings and impacts of your hypothesis better.

Reviewer #2 (Remarks to the Author):

This work analyses the impact of ocean changes on the observed increase of sea ice cover in the Ross Sea. The analysis is competent and generally supports the conclusions, and has the potential to be an extremely valuable and well-cited contribution to the field.

I have some concerns over aspects of the discussion, which is disappointing given the authors' deservedly-recognised expertise in this field. In particular the analysis does not sufficiently discuss competing hypotheses (i.e. wind forcing), and glosses over the importance of seasonality in places. Also, there is no mention of meridional transport and its impact on sea-ice-related freshwater fluxes.

The Ross Sea trends are largely at the ice edge, where there is a net melting of sea ice, suggesting that the feedback that is described here is to suppress melting rather than increase production; this is also consistent with work indicating that Ross sea ice trends are largely initialised in spring (Holland, 2014) - a discussion of these aspects would improve the paper.

There are also some minor methodological concerns. My specific comments are below - all of these concerns should be relatively easy to address, without further detailed analysis.

In general the writing is clear, but the ms is overly brief, with sense that a detailed rigorous analysis has been crammed into the short paper format. This is expressed in the rather weak discussion (alluded to above), but also some of the important analysis is tucked away in the Supplemental Material, although it isn't really 'supplemental' at all but is a major thread in the analysis. This makes the manuscript disjointed and difficult to follow. In places, the ms could do with explaining some of the core principles a little better for a general audience, particularly the relationship between sea ice and stratification.

1. Para 1, 5th sentence: I'm not sure that 'major contributor' is the best phrase here; do you mean the region with the largest observed expansion?

2. Para 1, final sentence: this is a strong conclusion, but I do not feel that the authors have done enough to address the hypothesis that trends are wind-forced (e.g. Haumann et al, 2014) to justify it.

3. Para 2, 3rd sentence: 'freshening' or 'freshwater input' would be better terms than '...modification of the hydrological cycle...'
4. Para 2 : I think you need to explain this physical process much better here. As written, it assumes that readers will be familiar that a) surface freshening increases stratification, b) increased stratification impedes the vertical transfer of heat, and c) in the Southern Ocean sea ice zone, CDW under the mixed layer is warm compared to the surface water.
5. Para 2 , final sentences do not seem consistent: first you say that increased stratification due to more sea ice leads to surface **cooling** and sub-mixed layer **warming**; then you invoke Schmidt et al's observation of **warming** WW in the Ross Sea, which is effectively the winter mixed layer. Surely that would reduce winter ice cover, but as you say the Ross Sea is unique in having a significant expansion throughout the year. There's a gap in this argument, possibly related to seasonality.
6. Para 3, 3rd sentence - you should explain how sea ice motion results in an effective freshwater transport. It would also be good to cite recent papers by Abernethy et al and Haumann et al
7. Para 3, 5th sentence: this sentence assumes that the reader is familiar with the seasonal evolution of the mixed layer under sea ice
8. No mention is made here, nor in the Methods or acknowledgements section, from where the BLUElink and OSISAF data were obtained.
9. 4th para, 3rd and 4th sentence: the model does not get the Amundsen Sea trends correct
10. 4th para, 4th sentence ; 'consistent', not 'consistently'
11. Figure 2 (and similarly supp. figure 2): The y-origin line is very faint here, barely visible. Also, panel indicates that model correlations are statistically-significant at all depths, even where the correlations are \sim zero. How can this be correct?
12. Para 6, 1st sentence: strictly speaking 'stratification' is related to the vertical gradient of density rather than salinity. A better phrasing might be '...and is amplified by, a salinity-related strengthening of the stratification and mixed-layer shoaling.'
13. Para 6, 2nd sentence: abbreviation MLD has not been defined.
14. Para 6, 2nd sentence: the changes in stratification/temperature described here for figure 3 are only true for regions of net sea ice melt, i.e. the ice edge and the Amundsen Sea. In regions of net ice production (ie the western end of the Ross Sea embayment) the very opposite pattern emerges. The discussion of the salinity/sea ice feedback mechanism is much diminished by ignoring its relationship to the horizontal transport of sea ice, and by implication freshwater.
15. Para 7, 6th sentence: Again, insufficient explanation: why is the positive feedback associated with year-round trends? This might be a good place to cite Holland, 2014; the described feedback seems to occur in regions of net melt, and Holland (2014) suggests that Ross trends are initiated in the melt season, so this might strengthen the authors' hypothesis
16. Para7, 8th sentence: I don't agree that using seasonal trends diminishes the data size such that trends/correlations are not robust. In my experience, autocorrelation of monthly sea ice data and the relationship between different seasons means that the true degrees-of-freedom of monthly data is not greatly less than that of year-to-year variability. If the authors have not accounted for this autocorrelation in their tests of significance, then that would be a

methodological flaw.

17. Para 6, final 2 sentences: I don't follow the reasoning here - how does a reversal in sign of the correlations '...demonstrate that the invoked mechanism is not specific to recent decades...?'

18. Para 8 and Supp Figure 4: The proposed feedback is largely related to sea ice melt and production, so why not use the melt/freeze seasons instead of the max/min ice cover seasons?

19. Para 8, 2nd sentence: it's not true that the trend in sea-ice-related FW flux (Supp Fig 4) has not spatial correlation to SIC trends. There is a change in the FW flux at the ice edge which clearly corresponds to the ASO SIC trend (Supp figure 3d), and changes in FW flux within the ice pack that correspond to summer SIC trends (Supp Figure 3b)

20. Para 9: the Rh value would only be ~ 1 if FW feedbacks were driving changes in the melt season. During freeze, entrainment by brine rejection invokes an upward ocean heat flux that means that a cooling of 3-5 times latent heat of freezing is required for a unit mass of sea ice (e.g. Martinson, 1990)

21. Para 9, reference 26: Wong and Riser (2011) might be a better ref here, since it is related specifically to the under-ice Argo programme

22. Para 10: again, some mention of seasonality, and differentiating melt and production for this feedback, would be appropriate.

23. Methods, Obs: is there a seasonal bias in the temporal distribution of profiles (I suspect there are many more in summer); what affect does this have on your annual averages and ultimately on your results?

24. Methods, Model: what is the DFS5.2 dataset and how might it be obtained?

25. Methods, diagnostics: How is statistical significance calculated, and is an adjustment made for temporal/spatial autocorrelation in the data? If not, why not?

26. Supp Material 1st para: similar to my comment about the seasonal distribution of ocean profiles, Figure S1 shows a spatial bias towards regions of net ice melt; what effect (if any) does this have? on the results? The profile of correlations may well be sensitive to the time of year that is represented.

REFERENCES

Abernathey, R. P., I. Cerovecki, P. R. Holland, E. Newsom, M. Mazloff, and L. D. Talley, 2016: Water-mass transformation by sea ice in the upper branch of the Southern Ocean overturning. *Nature Geosci*, advance online publication, 10.1038/ngeo2749

Haumann, F. A., D. Notz, and H. Schmidt, 2014: Anthropogenic influence on recent circulation-driven Antarctic sea ice changes. *Geophysical Research Letters*, 41, 8429-8437, 10.1002/2014GL061659.

Haumann, F. A., N. Gruber, M. Munnich, I. Frenger, and S. Kern, 2016: Sea-ice transport driving Southern Ocean salinity and its recent trends. *Nature*, 537, 89-92, 10.1038/nature19101.

Holland, P. R., 2014: The seasonality of Antarctic sea ice trends. *Geophysical Research Letters*, 41, 4230-4237, 10.1002/2014GL060172.

Martinson, D. G., 1990: Evolution of the Southern-Ocean Winter Mixed Layer and Sea Ice - Open Ocean Deep-Water Formation and Ventilation. *Journal of Geophysical Research-Oceans*, 95, 11641-11654, 10.1029/Jc095ic07p11641.

Wong, A. P. S., and S. C. Riser, 2011: Profiling Float Observations of the Upper Ocean under Sea Ice off the Wilkes Land Coast of Antarctica. *J Phys Oceanogr*, 41, 1102-1115, 10.1175/2011JPO4516.1.

Reviewer #3 (Remarks to the Author):

Review of: Antarctic sea-ice concentration trends sustained by vertical ocean heat redistribution (Lecomte et al)

"Antarctic sea-ice concentration trends sustained by vertical ocean heat redistribution" claims that positive sea ice trends in the Ross sea can set up an ocean-ice feedback mechanism that sustains sea ice growth and help explain observed trends in the Ross Sea over the satellite era. The argument is that a sufficiently large sea ice growth one year can lead to a large salinity flux through brine rejection into deeper layers that is not mixed into the upper layers in subsequent winters. Surface salinity export leads to increased stratification, resulting in a reduction in vertical oceanic heat fluxes (warming ocean waters at depth, cooling at the surface) and allowing for yet more ice growth.

This proposed mechanism itself is not new (e.g. it is presented in greater detail in Goosse and Zunz, 2014). What is new in this paper is analysis of both observations and a 20th Century ocean-ice hindcast simulation (ocean-ice model driven by atmospheric reanalysis data). This paper also suggests that this mechanism could help explain observed trends in the Ross Sea (the dominant driver of SH sea ice trends). Furthermore, this mechanism could also support that argument that observed trends are within natural variability.

This paper represents an important contribution to the ongoing questions – in particular why is Ross Sea sea ice (and thus Antarctic Sea Ice) increasing over the past 3 decades and how is it that models generally fail to replicate that trend? This research demonstrates from observations and model results that the ice-production-ocean-heat-storage presents a plausible explanations for regional sea ice increases.

This paper also presents a diagnostic (ratio of latent heat due to sea-ice volume trends and surface cooling trends to heat content changes in subsurface ocean) that could be quite useful in model analysis of SH sea ice.

I recommend the paper be published with some minor revisions outlined below.

The results are most applicable for the Ross Sea – the title and abstract however generalize to entire SH sea ice and I suggest making the title and/or abstract clearly state that results are primarily for the Ross Sea (although may have wider implications).

There are contradictions in years specified. Usually stated that trends in both observations and model are for 1979-2009, yet occasionally stated 1979-2013 (and from a reference dated 2008?) – clarify and make consistent.

It's not entirely clear how the trend correlations with depth are calculated. I think from model results that ocean temperatures, sea ice, etc are area-averaged over the given regions/domains then average correlations w.r.t. depth are calculated. I also think that for the observations the average is made from the available profiles (ones that meet criteria for inclusion as stated in the paper) within each region. I am not sure and this needs to be specified in methods and/or

supplement.

Include a regionally specific map of modeled sea ice trends in the Ross Sea in Figure 3 so that it is easier to see these compared with the other variables (rather than having to rotate, etc Figure 1).

Some discussion needs to be made of the apparent discrepancies in trends/correlations (1979-2009) in the Ross Sea region in Figure 3. For example, sea ice trends appear to be increasing throughout this area, particularly along the northern edge. Trends in ocean temperature at 150m however are not consistent even in the region of largest positive ice trends (northern edge). Why is this? How is it in a region where correlations should be positive in this region? I think if regional sea ice trends were shown it will be easier to see correlations with MLD, salinity gradient and most if not all of are of 150m ocean T trends – yet still a sentence or two explaining areas of apparent contradiction (e.g. northernmost edge in 150m T; western, inner Ross sea and all variables) would be valuable.

The observations in the Ross Sea are clearly much more spatially limited – could add to an argument that inner Ross Sea is not well represented by observations (and different processes likely at play in this region near the Ross Sea ice shelf and topographic wind forcing).

Reviewer #1

The authors use a comparison of an observational product and a model output to proof their hypothesis about the sea-ice extension origin.

The Manuscript is well written, and the findings are novel, though the authors do not test their hypothesis with different parameters or validate their results of modeled trends in ocean properties (e.g. Fig 3) against existing observational products, which are cited in their references. Furthermore the manuscript does need a more detailed description of the methods and data handling steps performed and an error analysis. Finally a discussion about the robustness of the results would be helpful, e.g. what other processes like the warming and rising of circumpolar deep water, could create a similar signal?

We thank the reviewer for her/his support. All issues are addressed in the answers (and associated modifications of the manuscript) to the specific comments below. In particular, the revised manuscript details the data analysis methods and contains extended discussion about the role of competing processes.

For a 2nd round of review, line numbers would be helpful,
Apologies for this, line numbers are now added.

In more detail:

Abstract, first sentence – there are quite strong regional differences, as the authors state themselves in the main text, this first sentence is to misleading, further more the reference to ‘global warming’ here is no appropriate, since to my knowledge it is not proven yet that the extended sea-ice in the southern ocean is linked to global warming.

Stating that the Ross Sea is the major contributor of sea-ice expansion needs a reference or rephrasing, sea ice drift patterns are not always simple to analyze or to be budgeted.

The first two sentences of the abstract were slightly reworded. However, we merely present some facts here: the sea-ice extent in the Southern Ocean has indeed been increasing for the past decades while our climate is warming. Those two facts are not proven to be related but rather represent a paradox, especially as compared to the dramatic sea-ice losses in the Arctic.

We no longer speak of the Ross Sea as “the major contributor to Antarctic sea-ice expansion”, but as “the region characterized by the largest and most significant expansion”, as shown by Parkinson et al. (2012) and Yuan et al. (2017).

1st sentence main text. ‘ice concentration’ - please make clear throughout the manuscript if you talk about concentration within a given, ‘fixed’ area, maximum sea ice extent (which concentration threshold) or sea ice area (subtracted open areas, areas with 50% coverage only counting as 50%...).

So far it is not always obvious to the reader what the authors do refer to.

We understand “ice concentration” as the fractional area of a reference observation or of a model grid cell covered by ice. It is now specified in the text and used only with this meaning.

reference 19, which shows the Winter Water warming also show a freshening of Winter Water – this would contradict your hypothesis of vertical ocean heat transfer, since it would lead to a salinification, wouldn’t it?

We agree, and realize the Winter Water trends may not be straightforward to interpret here, since the core depth of WW varies from 60 to 200 m (Fig 3c of Schmidt et al., 2014). The section was completely rewritten to simply explain the processes that may have influenced the Southern Ocean freshwater budget, including the potential contribution of recent anthropogenically-forced changes in winds.

You state that in an equilibrium climate ... salt rejected at depth one year is entrained again in the mixed layer the following year. – I doubt this to be true for the majority of the southern ocean due to pronounced zonal currents, westwards in the north, eastwards near the coasts. Please justify your assumption.

The sentence, as it was stated before, was indeed true in a one-dimensional situation only. We rephrased it and it now states: *“In an equilibrium climate, if we neglect for simplicity the potentially important contribution of horizontal transport, the situation is stable because the same amount of salt transported to depth one year is entrained in the mixed layer the following year.”*

Concerning your data set, the BlueLink Ocean Archive, I wonder why you restrict the analysis to 2013, since recent years significant amount of floats were available in your area of interest. – furthermore the BlueLink Ocean Archive is just binning data within a one degree grid cell. This method is not taking the current structures, bathymetry or data distribution in the southern ocean into account. Please justify the use of this data set in favor of the many other possible data sources? In particular on page 8 you state the availability of argo float and elephant seal data – this is publically available and can be used for your analysis!

The BlueLink Ocean Archive (BOA) offers several advantages: it is a publicly available, comprehensive, fully quality-controlled database. It provides temperature, salinity and mixed layer depth climatologies that include an estimate of the seasonal cycle for each mapped 0.5°x0.5° grid cell. These climatologies do account for bathymetry effects (see Dunn and Ridgway DSR 2002: Mapping ocean properties in regions of complex bathymetry). By providing Fourier coefficients for the first two harmonics describing the seasonal cycle, they allow computation of a climatological value for any day of year and any 0.5°x0.5° grid cell.

Following the Reviewer’s suggestion, we now use the most recent version of BOA. Nonetheless, as explained in the revised Methods section, because the OSISAF sea ice product only provides data up to April 2015, we restricted the analysis of observations to the period where year-round sea-ice concentration data is available: 1979-2014.

To avoid the aliasing of seasonal variability, we now compute trends in temperature, salinity and mixed layer depth from the anomalies with respect to the appropriate climatologies. Note that two BOA-based climatologies (called ‘CARS’) are available: one using all platforms (period 1900-2008); one using Argo only (period 2004-2014). After detailed examination of both climatologies, we concluded that the latter has the most reliable representation of the seasonal cycle and therefore used it. Data handling steps are described in detail in the Methods.

We considered using elephant seal data, which, indeed, could nicely complement the present database. However, because such profiles cover post-2004 years only, have a lower vertical resolution and slightly lower accuracy than other platforms, and are not incorporated in the CARS climatologies, they are not adequate for the computation of multi-decadal, depth-dependent temperature trends performed here. Consequently, to preserve consistency between profile data and climatologies, we chose not to incorporate elephant seal data in the present study.

For details on BOA and CARS, see:

<http://www.marine.csiro.au/~dunn/cars2009/>

<http://www.marine.csiro.au/~dunn/BOA.html>

https://researchdata.anders.org.au/bluelink-ocean-archive/692138?source=suggested_datasets

page 4 first paragraph: in the sentence 2 you contradict your statement from the abstract, where you emphasize that models fail to reproduce this. So which of these 2 sentences is correct?

This has been clarified in the revised version. Whereas fully coupled climate models reproduce Antarctic sea-ice trends over the last decades rather poorly, the ocean–sea-ice model forced by atmospheric reanalyses, which is employed here, does capture sea ice trends reasonably well. This difference can be largely explained by the fact that atmospheric reanalyses bear, by construction, the

information from observed sea-ice concentration fields. A forced ocean-ice model is thus more likely to reproduce the mean state and trends of sea ice than a fully coupled model.

Last sentence, same paragraph: what is a 'reasonable' number of observations, reasonable is spatial or temporal aspect? please rephrase that sentence.

We agree the sentence was inadequate. Our idea here is to justify our focus on the Ross Sea, where sea-ice changes are (1) the most important in magnitude, (2) the most significant statistically and (3) the most poorly understood so far in the whole Southern Ocean. We rephrased the last three sentences of this paragraph to put these elements forward.

page4 2nd paragraph: Please be more precise, which area, and grid cells do you compare, how do you compute the correlation, using monthly? Annual data? How do you handle data gaps that are common in observational data sets and how do you deal with the different temporal data coverage for each grid cell. Do you use the model only at the same grid cells as observations do exist?

Last with all the grid cells in the Ross Sea, you provide just one correlation curve, please add error bars to see the spread within the Ross Sea.

Trends are computed based on annual data (this is now specified in the corresponding paragraph of the new version of the manuscript). The other details are further discussed in the updated Methods section. However, as explained in the Methods section, we would like to stress that the correlations are not calculated over time. Instead, they are spatial correlations between trends in ice concentration and ocean temperature at depth. So sea-ice concentration and ocean temperature data are first averaged annually on a $1^\circ \times 1^\circ$ grid, and 1979-2014 trends are estimated independently for those two variables. We do so because the correlation signature presented in the paper appears only when looking at long-term trends. Then, the correlation between the spatial distribution of sea-ice concentration trends and that of ocean temperature trends at each depth is calculated. This explains why we end up with a single number for each depth, and hence a single depth-dependent correlation curve. Thus, no error bar or range of correlation can be added, but we do provide the depth ranges at which these correlations are significant. Please note that for calculating these correlations for observations, only grid cells including an estimate of both sea-ice concentration and ocean temperature trends are considered. For the model, we use all sea-ice covered grid cells.

page4 last paragraph: please guide the reader more thoroughly through your steps of conclusions. I found it hard to follow your chain of arguments.

Following your suggestion, the second part of this paragraph was rewritten.

page 5 1st paragraph: you state that the heat trapping is identified here in model and observation, please proof this hypothesis more carefully, e.g. by comparing salinity from observations in an identical way. For the limited region and limited data of the area, you need to back this bold statement, in particular because you state that other areas behave different.

Unfortunately, our diagnostic applied identically to salinity is inconclusive. It does not validate nor invalidate our hypothesis. Once all quality checks are applied to the observed trends, the final correlations are often very low and not significant. Here is the example of the Ross Sea:

Figure AR1: Ross Sea correlation coefficients between ice concentration trends and ocean salinity trends (1979-2014) as a function of depth, from observations.

However, we do provide new figures and discussion points. In particular, Figure 4 and the associated discussion address the present comment.

page 8 last paragraph: 'clearly' – please rephrase. – please consider rewriting the last paragraph, to explain the possible shortcomings and impacts of your hypothesis better.

As suggested, this paragraph has been rewritten and included in the "Outlook" section.

Reviewer #2

This work analyses the impact of ocean changes on the observed increase of sea ice cover in the Ross Sea. The analysis is competent and generally supports the conclusions, and has the potential to be an extremely valuable and well-cited contribution to the field.

I have some concerns over aspects of the discussion, which is disappointing given the authors' deservedly-recognised expertise in this field. In particular the analysis does not sufficiently discuss competing hypotheses (i.e. wind forcing), and glosses over the importance of seasonality in places. Also, there is no mention of meridional transport and its impact on sea-ice-related freshwater fluxes. The Ross Sea trends are largely at the ice edge, where there is a net melting of sea ice, suggesting that the feedback that is described here is to suppress melting rather than increase production; this is also consistent with work indicating that Ross sea ice trends are largely initialised in spring (Holland, 2014) - a discussion of these aspects would improve the paper.

There are also some minor methodological concerns. My specific comments are below - all of these concerns should be relatively easy to address, without further detailed analysis.

In general the writing is clear, but the ms is overly brief, with sense that a detailed rigorous analysis has been crammed into the short paper format. This is expressed in the rather weak discussion (alluded to above), but also some of the important analysis is tucked away in the Supplemental Material, although it isn't really 'supplemental' at all but is a major thread in the analysis. This makes the manuscript disjointed and difficult to follow. In places, the ms could do with explaining some of the core principles a little better for a general audience, particularly the relationship between sea ice and stratification.

We thank the reviewer for her/his support. We carefully considered all comments and modified the manuscript accordingly. The issues raised above are more extensively discussed (in new sections specifically dedicated to them, such as the influence of seasonality in our analysis and the role of the wind forcing and the meridional transport of freshwater in the studied mechanism). Some of the former Supplementary Material's content was brought back in the main manuscript, but some new elements were also added.

1. Para 1, 5th sentence: I'm not sure that 'major contributor' is the best phrase here; do you mean the region with the largest observed expansion?

Indeed, this was rephrased.

2. Para 1, final sentence: this is a strong conclusion, but I do not feel that the authors have done enough to address the hypothesis that trends are wind-forced (e.g. Haumann et al, 2014) to justify it.

We now use more cautious language and the role of wind forcing is discussed in more details in the section entitled "Competing hypotheses".

3. Para 2, 3rd sentence: 'freshening' or 'freshwater input' would be better terms than '...modification of the hydrological cycle...'

Agreed. The text has been modified accordingly.

4. Para 2 : I think you need to explain this physical process much better here. As written, it assumes that readers will be familiar that a) surface freshening increases stratification, b) increased stratification impedes the vertical transfer of heat, and c) in the Southern Ocean sea ice zone, CDW under the mixed layer is warm compared to the surface water.

This part was rewritten following the reviewer's advice.

5. Para 2 , final sentences do not seem consistent: first you say that increased stratification due to more sea ice leads to surface cooling and sub-mixed layer warming; then you invoke Schmidtke et al's observation of warming WW in the Ross Sea, which is effectively the winter mixed layer. Surely that would reduce winter ice cover, but as you say the Ross Sea is unique in having a significant expansion throughout the year. There's a gap in this argument, possibly related to seasonality.

We realize the argument was unclear. We completely rewrote the final part of this paragraph. By anticipation to the next comment below, the text now includes a discussion on the role of sea-ice advection in the effective freshwater transport.

6. Para 3, 3rd sentence - you should explain how sea ice motion results in an effective freshwater transport. It would also be good to cite recent papers by Abernethy et al and Haumann et al

This point is now addressed jointly with the modifications associated to comment 5 above. Besides, the discussion has been moved to the previous paragraph in the manuscript to clearly separate the wind-driven changes in freshwater transport from those caused by natural variability.

7. Para 3, 5th sentence: this sentence assumes that the reader is familiar with the seasonal evolution of the mixed layer under sea ice

A new sentence has been added to explain this point.

8. No mention is made here, nor in the Methods or acknowledgements section, from where the BLUElink and OSISAF data were obtained.

OSISAF is freely available online. This is now mentioned in the text and the reference was updated. We also provide the url where the BlueLink Ocean Archive can be accessed.

9. 4th para, 3rd and 4th sentence: the model does not get the Amundsen Sea trends correct

The sentences are now more cautious and mention the disagreement in the Amundsen Sea.

10. 4th para, 4th sentence ; 'consistent', not 'consistently'

The corresponding sentence has been removed because the whole section was rewritten.

11. Figure 2 (and similarly supp. figure 2): The y-origin line is very feint here, barely visible. Also, panel indicates that model correlations are statistically-significant at all depths, even where the correlations are \sim zero. How can this be correct?

We have enhanced the y-origin line in the revised version. The correlation significance is mainly controlled by our sample size. Here, with the model results, the spatial coverage is complete so significance is easily achieved even for very low correlation.

12. Para 6, 1st sentence: strictly speaking 'stratification' is related to the vertical gradient of density rather than salinity. A better phrasing might be '...and is amplified by, a salinity-related strengthening of the stratification and mixed-layer shoaling.'

Suggestion incorporated.

13. Para 6, 2nd sentence: abbreviation MLD has not been defined.

Now defined line 35.

14. Para 6, 2nd sentence: the changes in stratification/temperature described here for figure 3 are only true for regions of net sea ice melt, i.e. the ice edge and the Amundsen Sea. In regions of net ice production (ie the western end of the Ross Sea embayment) the very opposite pattern emerges. The discussion of the salinity/sea ice feedback mechanism is much diminished by ignoring its relationship to the horizontal transport of sea ice, and by implication freshwater.

The manuscript now includes two specific sections where the horizontal transport of sea ice is discussed, as suggested by the reviewer. Furthermore, we have added a figure showing the sea-ice

concentration trend (in the figure corresponding to figure 3 of the previous manuscript version), to relate more easily this field to other variables. We agree that the freshwater transport is important but we argue that vertical processes in the ocean play a central role in all the regions, as discussed in the section named “Competing hypotheses”.

15. Para 7, 6th sentence: Again, insufficient explanation: why is the positive feedback associated with year-round trends? This might be a good place to cite Holland, 2014; the described feedback seems to occur in regions of net melt, and Holland (2014) suggests that Ross trends are initiated in the melt season, so this might strengthen the authors' hypothesis

A dedicated discussion section related to seasonality and its impact on the feedback was included in the manuscript. The new reference was also added, as recommended.

16. Para7, 8th sentence: I don't agree that using seasonal trends diminishes the data size such that trends/correlations are not robust. In my experience, autocorrelation of monthly sea ice data and the relationship between different seasons means that the true degrees-of-freedom of monthly data is not greatly less than that of year-to-year variability. If the authors have not accounted for this autocorrelation in their tests of significance, then that would be a methodological flaw.

Concerning the tests of significance of our trends, we do test the null hypothesis and we do account for the autocorrelation of the model-based data, but it is not possible to do so for observations (please see the updated Methods section and the answer to the associated comment on significance tests below).

17. Para 6, final 2 sentences: I don't follow the reasoning here - how does a reversal in sign of the correlations '...demonstrate that the invoked mechanism is not specific to recent decades...'?

This section was rewritten and extended based on what was formerly in the Supplementary Material. To answer explicitly to the Reviewer's question, the correlation sign do not change from one period (1979-2009) to the other (1952-1978), because the feedback stays the same and can be applied both in case of an expansion or a decay of the ice cover. What changes is the sign in the trends. During the 1952-1978 period, the correlation signature of the feedback is present, but the ocean is cooling at depth at locations where sea-ice trends are mostly negative. The feedback occurs, but in this case amplifies the surface warming and sea ice decrease, which is what leads us to conclude that the signals we see are not typical only of the recent decades, but can also be identified in previous ones in model results.

18. Para 8 and Supp Figure 4: The proposed feedback is largely related to sea ice melt and production, so why not use the melt/freeze seasons instead of the max/min ice cover seasons?

This was indeed a good suggestion a priori. We tried to do so, but the opposition of signs in the trends from season to season, which is what we want to show, is less clear (see for instance Fig. AR2 below). Therefore, we preferred to keep the former figures, updated for the new periods of analysis.

Figure AR2: 1979-2013 seasonal trends in sea-ice concentration in the Southern Ocean, from the model simulations. AMJ and OND stand for April-May-June and October-November-December mean trends in ice concentration, respectively. These Spring and Autumn periods are defined as in Holland (2014).

19. Para 8, 2nd sentence: it's not true that the trend in sea-ice-related FW flux (Supp Fig 4) has not spatial correlation to SIC trends. There is a change in the FW flux at the ice edge which clearly corresponds to the ASO SIC trend (Supp figure 3d), and changes in FW flux within the ice pack that correspond to summer SIC trends (Supp Figure 3b)

Visually, it seems indeed that the two fields are spatially correlated (we mention this in the discussion), but the trends in freshwater flux associated with sea ice formation and melting display much larger spatial variability than trends in ice concentration do and the actual correlation between those fields is low. However, we discuss in more details in the revised version the mechanisms associated with northward transport of freshwater, underlining the potential importance of the latter in some regions.

20. Para 9: the R_H value would only be ~ 1 if FW feedbacks were driving changes in the melt season. During freeze, entrainment by brine rejection invokes an upward ocean heat flux that means that a cooling of 3-5 times latent heat of freezing is required for a unit mass of sea ice (e.g. Martinson, 1990) Martinson's (1990) feedback, which is different from the one presented here, acts on a seasonal time scale. A value corresponding to R_H can be defined for this feedback too and indeed its value would be very different from the one discussed here on interannual time scales. A remark was added to discuss this point, insisting also on the spatial variability of R_H .

21. Para 9, reference 26: Wong and Riser (2011) might be a better ref here, since it is related specifically to the under-ice Argo program.

We chose to remove the corresponding discussion from the main text, as it was deemed not central here.

22. Para 10: again, some mention of seasonality, and differentiating melt and production for this feedback, would be appropriate.

As explained above, a specific section was added to mention the impact of seasonality on the feedback and on its correlation signature.

23. Methods, Obs: is there a seasonal bias in the temporal distribution of profiles (I suspect there are many more in summer); what affect does this have on your annual averages and ultimately on your results?

Yes the distribution is biased towards summer. As explained above, we now work with profiles of ocean temperature anomaly in order to avoid aliasing of the seasonal cycle. This is discussed in the Methods section of the revised manuscript.

24. Methods, Model: what is the DFS5.2 dataset and how might it be obtained?

This information has been included in the revised version of the manuscript.

25. Methods, diagnostics: How is statistical significance calculated, and is an adjustment made for temporal/spatial autocorrelation in the data? If not, why not?

A description of the method used to test the significance was added in the revised version of the manuscript. For the model, we indeed account for the temporal autocorrelation of the data and make an adjustment in our significance tests. For the observations, it is however not done: the data distribution is originally very sparse and the lag-1 autocorrelation becomes insignificant. In this latter case, we just apply a regular Student t -test.

26. Supp Material 1st para: similar to my comment about the seasonal distribution of ocean profiles, Figure S1 shows a spatial bias towards regions of net ice melt; what effect (if any) does this have? On the results? The profile of correlations may well be sensitive to the time of year that is represented.

Regarding the spatial bias in the Ross Sea, indeed observations are available mostly in regions of net ice melt. This is a clear (but unavoidable) limitation for the observational estimate of correlations. However, Fig. 2a (observations) is very much consistent with Fig. 2c (model), both of which rely mostly on regions of net ice melt. This suggests a relative consistency between the observations and the model behaviour, at least in this area. Better sampling of the inner Ross Sea would allow a more complete and robust comparison across the different areas of the Ross Sea; this is not possible at present, unfortunately. Several sentences make note of the sampling biases, including in the paragraph dedicated to R_H .

Reviewer #3

“Antarctic sea-ice concentration trends sustained by vertical ocean heat redistribution” claims that positive sea ice trends in the Ross sea can set up an ocean-ice feedback mechanism that sustains sea ice growth and help explain observed trends in the Ross Sea over the satellite era. The argument is that a sufficiently large sea ice growth one year can lead to a large salinity flux through brine rejection into deeper layers that is not mixed into the upper layers in subsequent winters. Surface salinity export leads to increased stratification, resulting in a reduction in vertical oceanic heat fluxes (warming ocean waters at depth, cooling at the surface) and allowing for yet more ice growth.

This proposed mechanism itself is not new (e.g. it is presented in greater detail in Goosse and Zunz, 2014). What is new in this paper is analysis of both observations and a 20thCentury ocean-ice hindcast simulation (ocean-ice model driven by atmospheric reanalysis data). This paper also suggests that this mechanism could help explain observed trends in the Ross Sea (the dominant driver of SH sea ice trends). Furthermore, this mechanism could also support that argument that observed trends are within natural variability.

This paper represents an important contribution to the ongoing questions – in particular why is Ross Sea sea ice (and thus Antarctic Sea Ice) increasing over the past 3 decades and how is it that models generally fail to replicate that trend? This research demonstrates from observations and model results that the ice-production-ocean-heat-storage presents a plausible explanations for regional sea ice increases.

This paper also presents a diagnostic (ratio of latent heat due to sea-ice volume trends and surface cooling trends to heat content changes in subsurface ocean) that could be quite useful in model analysis of SH sea ice.

I recommend the paper be published with some minor revisions outlined below.

We thank the reviewer for her/his support. Her/His comments are addressed below.

The results are most applicable for the Ross Sea – the title and abstract however generalize to entire SH sea ice and I suggest making the title and/or abstract clearly state that results are primarily for the Ross Sea (although may have wider implications).

Agreed. The title and abstract were modified accordingly.

There are contradictions in years specified. Usually stated that trends in both observations and model are for 1979-2009, yet occasionally stated 1979-2013 (and from a reference dated 2008?) – clarify and make consistent.

This is fixed in the revised version, especially as the periods of analysis for the model and observations have changed in this new version of the manuscript.

It's not entirely clear how the trend correlations with depth are calculated. I think from model results that ocean temperatures, sea ice, etc are area-averaged over the given regions/domains then average correlations w.r.t. depth are calculated. I also think that for the observations the average is made from the available profiles (ones that meet criteria for inclusion as stated in the paper) within each region. I am not sure and this needs to be specified in methods and/or supplement.

The method section has been extended to clarify the way we computed our trends and correlations. For both datasets (observations and models), we first compute the annual fields. In the case of observations, an annual value is the mean of anomalies with respect to a climatological seasonal cycle for each grid cell. Once the annual values are computed, the trends are retrieved independently for all variables and for all available locations. Finally, the spatial correlation between the trends is evaluated.

It is precisely because we perform a spatial correlation between long-term trends that we end up with a single number at each depth in the main correlation diagnostic (Fig. 2). There is no averaging of correlations whatsoever.

Include a regionally specific map of modeled sea ice trends in the Ross Sea in Figure 3 so that it is easier to see these compared with the other variables (rather than having to rotate, etc Figure 1).

Thanks for the suggestion, which we followed.

Some discussion needs to be made of the apparent discrepancies in trends/correlations (1979-2009) in the Ross Sea region in Figure 3. For example, sea ice trends appear to be increasing throughout this area, particularly along the northern edge. Trends in ocean temperature at 150m however are not consistent even in the region of largest positive ice trends (northern edge). Why is this? How is it in a region where correlations should be positive in this region? I think if regional sea ice trends were shown it will be easier to see correlations with MLD, salinity gradient and most if not all of are of 150m ocean T trends – yet still a sentence or two explaining areas of apparent contradiction (e.g. northernmost edge in 150m T; western, inner Ross sea and all variables) would be valuable.

We agree that interpreting our results was difficult, for two reasons. The first is that, indeed, a regional sea-ice concentration trend map was missing for comparison. The second is that the temperature trends were shown at a constant depth, while the warming occurs at different depths depending on the mean state of the MLD. We now show the trends in temperature and in salinity gradient based on the depth of the maximum ocean temperature trend in the first 300 m below the sea surface (see Fig. 3 and Supplementary Fig. 3). As a result, the salinity stratification and subsurface warming are apparent in most of the Ross Sea.

The relatively strong cooling at depth in the northernmost part of the sea-ice zone and the changes observed near the ice shelf are unrelated to sea-ice trends, which are very weak in these areas. North of the ice edge, the temperature change is characterized by a homogeneous cooling in and below the mixed layer. Near the coast, both the cooling and deepening of the mixed layer likely result from processes taking place locally due to the presence of the ice shelf and katabatic winds, as suggested by the Reviewer in the last comment below.

The observations in the Ross Sea are clearly much more spatially limited – could add to an argument that inner Ross Sea is not well represented by observations (and different processes likely at play in this region near the Ross Sea ice shelf and topographic wind forcing).

The fact that the inner Ross Sea specifically is not well-sampled is now stated in the paragraph dedicated to R_H .

References

Dunn, J. R., & Ridgway, K. R. Mapping ocean properties in regions of complex topography. *Deep Sea Research Part I: Oceanographic Research Papers*, 49(3), 591-604 (2002).

Holland, P. R. The seasonality of Antarctic sea ice trends. *Geophysical Research Letters* 41, 4230–4237 (2014).

Parkinson, C. L. & Cavalieri, D. J. Antarctic sea ice variability and trends, 1979–2010. *The Cryosphere* 6, 871–880 (2012).

Yuan, N., Ding, M., Ludescher, J. & Bunde, A. Increase of the Antarctic Sea Ice Extent is highly significant only in the Ross Sea. *Scientific Reports* 7 (2017).

Reviewers' comments:

Reviewer #2 (Remarks to the Author):

The manuscript is much improved and the arguments are made more clearly. However, there are still some aspects that require further explanation, and some aspects where the supplemental material is inconsistent with the manuscript.

Specific Comments:

line 56: the authors cite Yuan et al (2017) to assert that trends are only significant in the Ross Sea, but in the supplemental material line 15 assert that due to seasonality, trends based on annual values of SIC may be meaningless; but this was exactly the flawed approach used by Yuan et al.

Figure 2b and Supplemental figure 5d: the authors claim that because the spatial correlations between SIC trend and ocean temperature trend are robust across different time periods (1979-2013 vs 1952-1978) the feedback they describe does not require external forcing. However, the very different correlation profiles for the Ross Sea in these figures are not consistent with this argument.

line 74-76: it would be useful to explicitly state that the +ve correlation at depth implies ocean heat trapped below the mixed layer. This is stated later in line 84 but would be helpful here.

line 130-131: Why does your feedback suggest trends that are of the same sign throughout the year? you haven't explicitly stated this.

line 161 typo: '...freswhater...'

line 183-185: what's the initial perturbation causing this trend, if not the wind? How do you know it's not just model drift in the ocean temperature? In general this section is incomplete without some idea of what is causing the trends in figure 5, in the absence of any trend in the boundary forcings of the simulation

lines 206-211. This argument seems incomplete also, because it's only discusses MLD anomalies in the melt season. What if the shoaling happened in freeze? You'd get more ice, more brine rejection, and it would cancel out the shoaling anomaly (a -ve rather than +ve feedback). Again, the impact of this feedback on the freeze vs melt seasons is not adequately addressed.

Methods, line 278-279: the authors have addressed my comments about temporal autocorrelation, but not about spatial autocorrelation, in their profiles of correlation between model SIC/ocean temperature trend. The resolution of NEMO-LIM3.6 is not specified here, but I expect that there would be some spatial autocorrelation across the model grid cells.

Reviewer #3 (Remarks to the Author):

This second version of the paper is much improved and the concerns in my first review have been addressed. Thank you also for including line numbers for reference. I have a couple minor recommendations/questions:

General

The shading of significant areas (gray shading) to show depths at which correlations are significant I find a bit confusing - or perhaps I am misunderstanding - in figures 2b, 5a, and supplementary material figures 2, 5. The significance test (95% confidence) should be indicating which

correlations exceed the minimum for the sample size that exceed (absolute value) the 95% confidence limit however the white area (which should show regions that are not significant) falls at depths where the absolute value of the correlation coefficient is not at a minimum. So either the sample sizes at these depths is much smaller, or the gray shading isn't aligned properly (I suspect the latter?).

Lines 106-107

The only sector that does not have a positive correlation between sea ice trends and subsurface temperature trends in the early time period (1952-1978) is the Ross Sea? Why? The primary focus of the paper is the Ross Sea so some mention of why you think this might be is warranted...are these relationships b/w sea ice concentration trends and ocean temperature at depth not robust in time? Why would they change?

Specific

Lines:

14

change "Ross Sea" to "Ross and Weddell Seas". Although Ross is clearly largest contributor to late 20th C sea ice trends, Weddell is a not so distant 2nd (and clearly important in the observations)

59

change "Amundsen" to "Weddell"

Amundsen looks ok to me, Weddell not so much (at least what I can see).

64

verb tense – change "determine" to "determining"

113

"surface warming" – surface temperature trends are not shown in supplementary Figure 6 so I think you mean "salinity gradients"

144

change "warming layer" to "warming subsurface layer"

192

Suggest starting a new paragraph here.....

Supplementary Figure 6

Change "ocean surface properties" to "upper ocean properties"

Reviewer #2

The manuscript is much improved and the arguments are made more clearly. However, there are still some aspects that require further explanation, and some aspects where the supplemental material is inconsistent with the manuscript.

Specific Comments:

line 56: the authors cite Yuan et al (2017) to assert that trends are only significant in the Ross Sea, but in the supplemental material line 15 assert that due to seasonality, trends based on annual values of SIC may be meaningless; but this was exactly the flawed approach used by Yuan et al.

We have removed the unnecessary reference to the results of Yuan et al. (2017) (see line 48).

Figure 2b and Supplemental figure 5d: the authors claim that because the spatial correlations between SIC trend and ocean temperature trend are robust across different time periods (1979-2013 vs 1952-1978) the feedback they describe does not require external forcing. However, the very different correlation profiles for the Ross Sea in these figures are not consistent with this argument.

The Ross Sea is the only sector which does not exhibit the positive subsurface correlation over 1952-1978. Evidence from all other sectors does support the presence of a vertical reorganization of energy. Furthermore, Supplementary Figure 6 shows that, associated with the simulated 1952-1978 Ross Sea retreat of sea ice, mixed layer deepening, stratification decrease and deep (150 m) cooling are simulated over much of the sector. Opposite trends at the gyre centre, where the mixed layer is anomalously shallow, explain the absence of a clear overall signature of the feedback (Supplementary Figure 5d). The discrepancy is thus relatively local and therefore does not discard the general validity of our statement: in the model, the ice-ocean feedback operates over most of the Antarctic sea ice zone during 1952-1978, a period when anthropogenic forcing was significantly weaker than during 1979-2014.

We added a note to explain the absence of a clear feedback signature in the Ross Sea in this earlier time period (lines 111-114): *“Supplementary Fig. 6 shows a deepening of the mixed layer and cooling of subsurface waters co-occurring with negative sea-ice concentration trends and stratification weakening over much of the Antarctic sea ice zone (note that the situation is different at the centre of the Ross gyre due to a shallow MLD anomaly, blurring the feedback signature when averaging over the Ross sector over that period)”*.

line 74-76: it would be useful to explicitly state that the +ve correlation at depth implies ocean heat trapped below the mixed layer. This is stated later in line 84 but would be helpful here.

We followed the suggestion. The sentences now read (lines 72-76): *“As expected, ice concentration and sea surface temperature trends are negatively correlated, meaning that an increase (decrease) in ice concentration is associated with a surface cooling (warming). Nonetheless, the correlation becomes positive as depth increases and reaches a maximum at a depth of nearly 120 m in observations and 150 m in the model: this entails that an increase (decrease) in ice concentration is associated with heat gain (loss) at depth.”*

line 130-131: Why does your feedback suggest trends that are of the same sign throughout the year? you haven't explicitly stated this.

We clarified the paragraph (lines 133-135). The feedback favors the production of ice when heat is stored at depth (and a reduced ice production or faster melting when heat stored at depth is released). Hence, it cannot explain alone opposite trends across seasons.

line 161 typo: '...freswhater...'

Thank you for pointing this out.

line 183-185: what's the initial perturbation causing this trend, if not the wind? How do you know it's not just model drift in the ocean temperature? In general this section is incomplete without some idea of what is causing the trends in figure 5, in the absence of any trend in the boundary forcings of the simulation

It is difficult to answer this question precisely, as the initial perturbation could be anything, from a temporary anomaly in the boundary conditions to a thermodynamical perturbation, as shown in Goosse and Zunz (2014). A note was added at the end of the corresponding discussion (lines 194-197).

lines 206-211. This argument seems incomplete also, because it's only discusses MLD anomalies in the melt season. What if the shoaling happened in freeze? You'd get more ice, more brine rejection, and it would cancel out the shoaling anomaly (a -ve rather than +ve feedback). Again, the impact of this feedback on the freeze vs melt seasons is not adequately addressed.

We made the argument clearer by indicating that a “year-round” mixed-layer shoaling is assumed (line 215). Brine rejection during freezing will be more effective at deepening a shallower mixed layer, but it is assumed that it will not be able to compensate totally for the initial negative anomaly. Furthermore, in regions of net annual surface freshwater gain, yearly precipitation and seasonal ice melt, which are there larger than brine rejection, will then lower the mixed-layer salinity below its initial level. The resulting stratification increase consolidates, and possibly amplifies, the negative MLD anomaly. The same scenario repeats over the following year.

We agree that brine rejection during freezing may (or may not) cancel a negative mixed-layer anomaly, but, in this respect, the presented mechanism can simply be seen as a long-term modulation of Martinson (1990)’s feedback.

Methods, line 278-279: the authors have addressed my comments about temporal autocorrelation, but not about spatial autocorrelation, in their profiles of correlation between model SIC/ocean temperature trend. The resolution of NEMO-LIM3.6 is not specified here, but I expect that there would be some spatial autocorrelation across the model grid cells.

The model data is indeed spatially autocorrelated, but it is difficult to account for this autocorrelation in our diagnostic, in a formal way at least. With respect to temporal autocorrelation, a red noise model is assumed and a standard procedure is applied using the lag-1 autocorrelation of the time series to compute the effective sample sizes and assess the significance of the trends. For the spatial autocorrelation, it is not as straightforward as there is no predetermined model to use in order to reduce the sample size in a similar way.

Nonetheless, in order to address your concern, we made a simple test, by producing Fig. 2b after artificially reducing the sample size (see Fig. AR1 below). If we consider that any grid cell is correlated with its direct neighbours in every directions, we can select only 1 out of every 9 cells in the model grid and consequently reduce the sample size by a factor of 9 to compute the correlations. Although quantitatively different, Fig. AR1 conveys the same message as the original Fig. 2b, which supports that spatial autocorrelation does not invalidate or change our results. For this reason and because this test is not a formally clean way to account for spatial autocorrelation in our results, we maintain Fig. 2b in its current state. A note about this is added lines 287-291. Performing the same test with observation-based data is however difficult, since the data are sparse and quantitatively limited. An alternative way to check the spatial autocorrelation in the observational data could have been to estimate the temporal correlation of temperature time series between grid cells, but, once again, such a robust correlation is not easily obtained as observations in different places are not necessarily available at the same time.

Figure AR1: Correlation coefficients between ice concentration and depth-dependent ocean temperature trends in the Ross Sea, based on model data. Grey areas show the depths at which the correlations are significant at the 95% confidence level. Correlations are computed after artificial reduction of the sample size by a factor of 9 (i.e., 1 out of every 9 significant trends is selected to compute the correlations), compared to Fig. 2b in the main manuscript. The significance of trends itself is assessed by testing the null hypothesis with a Student's *t*-test, accounting for temporal autocorrelation in the time series (see Methods).

Reviewer #3

This second version of the paper is much improved and the concerns in my first review have been addressed. Thank you also for including line numbers for reference. I have a couple minor recommendations/questions:

General

The shading of significant areas (gray shading) to show depths at which correlations are significant I find a bit confusing - or perhaps I am misunderstanding – in figures 2b, 5a, and supplementary material figures 2, 5. The significance test (95% confidence) should be indicating which correlations exceed the minimum for the sample size that exceed (absolute value) the 95% confidence limit however the white area (which should show regions that are not significant) falls at depths where the absolute value of the correlation coefficient is not at a minimum. So either the sample sizes at these depths is much smaller, or the gray shading isn't aligned properly (I suspect the latter?).

These correlations (e.g., around ~130 m in Fig. 2b & 5a) are non-significant simply because the data at these depths are most auto-correlated (temporally). Therefore, many trends at these depths are non-significant and the sample size for the correlation calculations is indeed drastically reduced.

Lines 106-107

The only sector that does not have a positive correlation between sea ice trends and subsurface temperature trends in the early time period (1952-1978) is the Ross Sea? Why? The primary focus of the paper is the Ross Sea so some mention of why you think this might be is warranted...are these relationships b/w sea ice concentration trends and ocean temperature at depth not robust in time? Why would they change?

The Ross Sea is the only sector which does not exhibit the positive subsurface correlation over 1952-1978. Evidence from all other sectors does support the presence of a vertical reorganization of energy. Furthermore, Supplementary Figure 6 shows that, associated with the simulated 1952-1978 Ross Sea retreat of sea ice, mixed layer deepening, stratification decrease and deep (150 m) cooling are simulated over much of the sector. Opposite trends at the gyre centre, where the mixed layer is anomalously shallow, explain the absence of a clear overall signature of the feedback (Supplementary Figure 5d). The discrepancy is thus relatively local and therefore does not discard the general validity of our statement: in the model, the ice-ocean feedback operates over most of the Antarctic sea ice zone during 1952-1978, a period when anthropogenic forcing was significantly weaker than during 1979-2014.

We added a note to explain the absence of a clear feedback signature in the Ross sea in this earlier time period (lines 111-114): *“Supplementary Fig. 6 shows a deepening of the mixed layer and cooling of subsurface waters co-occurring with negative sea-ice concentration trends and stratification weakening over much of the Antarctic sea ice zone (note that the situation is different at the centre of the Ross gyre due to a shallow MLD anomaly, blurring the feedback signature when averaging over the Ross sector over that period)”*.

Specific

Lines:

14

change “Ross Sea” to “Ross and Weddell Seas”. Although Ross is clearly largest contributor to late 20th C sea ice trends, Weddell is a not so distant 2nd (and clearly important in the observations)

Suggestion included (line 14).

59

change “Amundsen” to “Weddell”

Amundsen looks ok to me, Weddell not so much (at least what I can see).

We now state: "Except for the outer Weddell and inner Amundsen Seas" (line 51).

64

verb tense – change "determine" to "determining"

Correction included (line 57).

113

"surface warming" – surface temperature trends are not shown in supplementary Figure 6 so I think you mean "salinity gradients"

The sentence was corrected (lines 111-113) and now reads: *"Supplementary Fig. 6 shows a deepening of the mixed layer and cooling of subsurface waters co-occurring with negative sea-ice concentration trends and stratification weakening over much of the Antarctic sea ice zone"*.

144

change "warming layer" to "warming subsurface layer"

Changed as suggested (line 149).

192

Suggest starting a new paragraph here.....

Done (now line 198).

Supplementary Figure 6

Change "ocean surface properties" to "upper ocean properties"

Changed as suggested.

REVIEWERS' COMMENTS:

Reviewer #2 (Remarks to the Author):

The authors have largely addressed my concerns, with the exception of the following point, pertaining to trends in the model run forced by climatological winds:

My original comment:

line 183-185: what's the initial perturbation causing this trend, if not the wind? How do you know it's not just model drift in the ocean temperature? In general this section is incomplete without some idea of what is causing the trends in figure 5, in the absence of any trend in the boundary forcings of the simulation

Authors' response:

It is difficult to answer this question precisely, as the initial perturbation could be anything, from a temporary anomaly in the boundary conditions to a thermodynamical perturbation, as shown in Goosse and Zunz (2014). A note was added at the end of the corresponding discussion (lines 194-197).

If you don't know what drives the trend, then how do you know the temperature profile changes aren't just model drift, and the sea ice isn't just responding to that? That wouldn't be a feedback, but a sea ice change forced by ocean temperature changes.

I have to say, in the absence of a further explanation of this trend, drift seems the most likely explanation

In short, I still don't think that this part of the study fulfils the aim of being further proof of the authors' hypothesis, and would question its validity without further analysis

I would recommend publication on the authors addressing my final point, either by removing that part of the paper (pertaining to Figure 5), or being able to prove me wrong! (i.e. that it's a feedback process and not just drift in the ocean model).

Final revisions – Answer to the Reviewer

Below the comment from Reviewer #2 (in black), and our answer (in blue).

“The authors have largely addressed my concerns, with the exception of the following point, pertaining to trends in the model run forced by climatological winds:

My original comment:

line 183-185: what's the initial perturbation causing this trend, if not the wind? How do you know it's not just model drift in the ocean temperature? In general this section is incomplete without some idea of what is causing the trends in figure 5, in the absence of any trend in the boundary forcings of the simulation

Authors' response:

It is difficult to answer this question precisely, as the initial perturbation could be anything, from a temporary anomaly in the boundary conditions to a thermodynamical perturbation, as shown in Goose and Zunz (2014). A note was added at the end of the corresponding discussion (lines 194-197).

If you don't know what drives the trend, then how do you know the temperature profile changes aren't just model drift, and the sea ice isn't just responding to that? That wouldn't be a feedback, but a sea ice change forced by ocean temperature changes.

I have to say, in the absence of a further explanation of this trend, drift seems the most likely explanation

In short, I still don't think that this part of the study fulfils the aim of being further proof of the authors' hypothesis, and would question its validity without further analysis

I would recommend publication on the authors addressing my final point, either by removing that part of the paper (pertaining to Figure 5), or being able to prove me wrong! (i.e. that it's a feedback process and not just drift in the ocean model).”

Model drift as a driver for these trends is excluded, as for instance the trends have opposite signs in the Ross Sea over the two periods, 1952-1978 and 1979-2009. However, we recognize that the discussion, formerly lines 195-197 (now 238-241 in track changes version), is speculative in the absence of further details. Following the recommendation from Reviewer #2 and the Editor, we therefore

removed that part of the text. Additionally, to clarify our setup for the experiment driven by climatological winds, we amended line 197 of the manuscript (line 231 of the track changes version) to specify that only the wind is climatological and that all other forcing fields are left unchanged compared to the main simulation of the study.